Leaf water potential of field crops estimated using NDVI in ground-based remote sensing—opportunities to increase prediction precision

Dong Xuejun 1 xuejun.dong@ag.tamu.edu
Peng Bin 2
Sieckenius Shane 1
http://orcid.org/0000-0003-4514-0042 Raman Rahul 1 3
http://orcid.org/0000-0001-5221-7576 Conley Matthew M. 4
Leskovar Daniel I. 1
1 Texas A&M AgriLife Research and Extension Center at Uvalde , Uvalde, TX , United States
2 Yancheng Institute of Technology , Yancheng City, Jiangsu , China
3 Department of Soil and Crop Sciences, Texas A&M University , College Station, TX , United States
4 USDA-ARS, U.S. Arid-Land Agricultural Research Center , Maricopa, AZ , United States
Yu Le
Electronic publication date: 2021 Aug 18
Publication date: 2021
Volume: 9
Electronic Location ID: e12005
Received 2021 Mar 9; Accepted 2021 Jul 27
Copyright year: 2021
License: This is an open access article, free of all copyright, made available under the Creative Commons Public Domain Dedication. This work may be freely reproduced, distributed, transmitted, modified, built upon, or otherwise used by anyone for any lawful purpose.
License URL: https://creativecommons.org/publicdomain/zero/1.0/

Keywords: Drought stress, Leaf water status, Statistical resampling, Multispectral sensor, Ordinary least-squares, Proximal sensing, Weighted least-squares, Measurement errors, Diurnal change, Model coefficients

Funding: Texas A&M AgriLife Research Cropping System Project USDA NIFA J. H. Biotech Inc Jiangsu Overseas Scholar Program The push-wheel phenotyping cart was built using funds from the Texas A&M AgriLife Research Cropping System Project entitled “Root-shoot phenotyping and water balance characterization to improve water use efficiency and productivity of cropping systems in Texas” (Xuejun Dong and Daniel I. Leskovar). The work was also supported by the Texas A&M AgriLife Research Hatch project TEX09574 funded by the USDA NIFA (Xuejun Dong), J. H. Biotech Inc. project “Field test of SYNERGIZER 8-32-4 and SOLUPHOS SP in improving phosphorus nutrition and drought tolerance in cotton” (Daniel I. Leskovar and Xuejun Dong), and Jiangsu Overseas Scholar Program for University Prominent Young & Mid-Career Teachers and Presidents (Bin Peng). The funders had no role in study design, data collection and analysis, decision to publish, or preparation of the manuscript.

==============================
Remote-sensing using normalized difference vegetation index (NDVI) has the potential of rapidly detecting the effect of water stress on field crops. However, this detection has typically been accomplished only after the stress effect led to significant changes in crop green biomass, leaf area index, angle and position, and few studies have attempted to estimate the uncertainties of the regression models. These have limited the informed interpretation of NDVI data in agricultural applications. We built a ground-based sensing cart and used it to calibrate the relationships between NDVI and leaf water potential (LWP) for wheat, corn, and cotton growing under field conditions. Both the methods of ordinary least-squares (OLS) and weighted least-squares (WLS) were employed in data analysis, and measurement errors in both LWP and NDVI were considered. We also used statistical resampling to test the effect of measurement errors of LWP on the uncertainties of model coefficients. Our data showed that obtaining a high value of the coefficient of determination did not guarantee a high prediction precision in the obtained regression models. Large prediction uncertainties were estimated for all three crops, and the regressions obtained were not always significant. The best models were obtained for cotton with a prediction uncertainty of 27%. We found that considering measurement errors for both LWP and NDVI led to reduced uncertainties in model coefficients. Also, reducing the sample size of LWP measurement led to significantly increased uncertainties in the coefficients of the linear models describing the LWP-NDVI relationship. Finally, potential strategies for reducing the uncertainty relative to the range of NDVI measurement are discussed.

Introduction

The accurate determination of leaf water potential (LWP) of field crops plays an important role in agricultural irrigation scheduling (Kramer, 1988; Jones, 2004), regional crop health surveys (Pu et al., 2004; Maciel et al., 2020), and in the screening of drought tolerance of crop genotypes for use in arid and semi-arid regions (Kumar & Singh, 1998). Yet traditional methods using the leaf pressure chamber or hygrometer are destructive, time-consuming, and not efficient enough for quantifying a large number of crop genotypes (Mart, Veneklaas & Bramley, 2016). This challenge may be overcome by the application of modern remote-sensing techniques (Jones & Vaughan, 2010; Gago et al., 2015; Ihuoma & Madramootoo, 2017). The normalized difference vegetation index (NDVI) derived from low-cost sensors measuring reflectance spectrum in the visible (VIS) and near infrared (NIR) bands, has been traditionally used mainly as a course ‘greenness’ index, to indicate canopy cover and plant vigor (Rouse et al., 1974; Carlson & Ripley, 1997). Recently, NDVI has been used to infer the effect of water availability and water stress in plants (Irmak et al., 2011; Aguilar et al., 2012; Hunink et al., 2015; Bronson et al., 2017; Hunsaker et al., 2007). NDVI was also used to characterize changes in shoot water content of sesame during drydown (Dong, Feng & Zemach, 2021). For most instances, however, what NDVI actually captures is most likely the accumulated effect of changed leaf water status on plant growth, as indirectly indicated by structures involving water such as leaf area index, or green biomass, instead of the leaf water potential directly. Waiting until the stress effect has led to significant changes in crop biomass and generic NDVI detection may be too late to guide timely crop management, such as irrigation scheduling. Excluding this confounding effect regarding changes in plant biomass, as well as the confounding effects due to environmental conditions, Elsayed, Mistele & Schmidhalter (2011) successfully demonstrated the direct linkage between LWP and spectral reflectance obtained from the VIS and NIR bands for wheat and maize. They used climate chambers to only allow light intensities to change in the short-term of 2–3 h while keeping other factors, such as humidity and temperature, unchanged. This allowed them to observe the rapid reduction in LWP during the early stage of leaf water loss when the change in leaf water content may or may not show up, as seen from the concept of the pressure-volume relations during leaf water loss (Cheung, Tyree & Dainty, 1975). The biophysical mechanisms underlying the changes in vegetation indices in relation to water stress lie in the changes in leaf reflectance from the VIS and NIR bands. The reduced NDVI due to water stress may be caused by the increased red band reflectance associated with decreased chlorophyll content or absorption (Carter, 1993), or alternatively, it may originate from the reduced NIR reflectance associated with reduced cell wall-air interfaces (Grant, 1987) or changes in morphology, such as reduced leaf thickness with the reduction of LWP (Syvertsen & Levy, 1982; Knapp & Carter, 1998; Slaton, Hunt & Smith, 2001).

The strong and direct linkage between the observed changes in LWP and the concomitant changes in selected spectral indices (Elsayed, Mistele & Schmidhalter, 2011) calls for further applications to survey leaf water status for a large number of crop genotypes, as well as under field conditions where the rapid change in light condition is the main factor influencing leaf water potentials. Several other researchers investigated the feasibility of remotely detecting leaf/canopy water status using the airborne or ground-based sensing tools accompanied with LWP measurement (Stimson et al., 2005; Mastrorilli et al., 2010; Baluja et al., 2012; Zarco-Tejada, González-Dugo & Berni, 2012; Maciel et al., 2020). By precisely measuring needle spectral properties using a portable spectrometer, Stimson et al. (2005) determined that NDVI captured the water potential signals of needles of Pinus edulis (R2 = 0.35), but not those of Juniperus monosperma. The reason of the difference was presumably due to the reduced reflectance and albedo caused by the denticulate leafy shoots of J. monosperma as opposed to the straight needles of P. edulis that allowed them to be tightly packed and measured by the spectrometer. Baluja et al. (2012) found that both the selected spectral indices such as NDVI and thermal indices were highly effective in detecting the variability of LWP of a rain-fed vineyard due to natural variability of soil properties. Yet, the correlation between LWP of a citrus orchard and NDVI was weaker (R2 = 0.24) than that with crown temperature or chlorophyll fluorescence calculated from a micro-hyperspectral imagery (Zarco-Tejada, González-Dugo & Berni, 2012). Field measurements spanning multiple years in a coffee plantation (Maciel et al., 2020) documented the usefulness of NDVI in remote estimation of LWP of coffee plants both in the dry season and rainy season (R2 = 0.82). However, Mastrorilli et al. (2010) cautioned about the operative use of NDVI in irrigation scheduling for tomato, due to the narrow range of the NDVI signals as well as “errors innate to the measuring technique”.

Despite the comprehensive nature of the above cited works (Zarco-Tejada, González-Dugo & Berni, 2012), the field-measured data points of LWP were almost always sparse, as compared with those from the remote sensing platforms, and very few of the above-mentioned studies provided estimates of errors or uncertainties of the regression coefficients or those of the predictions by the obtained regression models (but see Rallo et al., 2014). Because both NDVI and LWP are measured values, they both carry statistical uncertainties. Ignoring them in the data analysis may lead to missed opportunities to identify the dominant factors influencing the prediction precision of the models, thus affecting the informed interpretation of NDVI data in agricultural and biological application. In the case of remote sensing of LWP in particular, a narrow response range of NDVI might suggest large relative errors; thus, the practical implication of having a higher or a lower value of coefficient of determination can better be assessed by considering error propagation from the independent to the dependent variables. A fuller account of measurement errors can be found in the description of physical systems (Taylor, 1982; Bevington & Robinson, 2003), and readily usable routines have been developed (Reed, 1989, 1992; Press et al., 1992; Cantrell, 2008; Reed, 2010, 2015) but less utilized to analyze biological data.

As mentioned above, a major limiting factor for calibrating the LWP-NDVI relationship for field crops is in acquiring sufficient number of measurements of LWP using one of the accepted reference techniques. For example, using the pressure chamber, about 3–6 leaf samples per treatment are typically employed to measure LWP (Mastrorilli et al., 2010; Baluja et al., 2012). When there are more samples to measure within a limited time frame, the sample size of LWP may be reduced. But it is unknown how an increased error due to reduced sample size of LWP would affect the uncertainties of the model coefficients of the LWP-NDVI relationship? Answers to this question are critical for improving the efficiency by which the LWP-NDVI relationship is calibrated, but have rarely been addressed in the literature.

Ground-based sensing platforms have been developed to assist in field crop phenotyping (White & Conley, 2013; Bai et al., 2016; Barker et al., 2016; Zhang et al., 2019). Murray et al. (2016) developed an automated, high-clearance phenotyping vehicle to accommodate tall crops such as corn. In this study, the utility of the sensing cart design of White & Conley (2013) was improved upon, and extended by adding a backpack sensing frame that can be carried by one person to survey tall crops. To minimize the confounding effects of changed biomass or green leaf area, LWP and NDVI were measured concurrently at different times of a day for different crops under field conditions. The objectives of this study were to: Apply and test a push-wheel sensing cart for phenotyping field crops of wheat, corn and cotton;

Develop the regression equations describing LWP as a function of NDVI for different crops by considering measurement errors in both LWP and NDVI; and

Test the effect of measurement errors of LWP on the prediction precision of the obtained models.

Materials and Methods

Application and construction of a ground-based phenotyping tool

To assist in field crop phenotyping efficiently and in a consistent manner, we developed a portable proximal sensing cart based on the model of White & Conley (2013). We added wheel assembly improvements to the cart frame, facilitating easy maneuverability by one person in field conditions, as well as adding nested threaded clamps to adjust the cart width, allowing crop measurement with different row spaces (Fig. 1). Attached to the cart are three highly accurate canopy temperature sensors, one ultrasonic crop displacement sensor, one active light NDVI sensor, and a marine grade GPS receiver (A–F in Fig. 2). We also constructed a backpack frame that allows selected measurement of NDVI in tall crops, such as corn, sorghum and sesame (G in Fig. 2). The ACS-430 Crop Circle sensor measures canopy reflectance from three optical channels (red–670 nm, red-edge–730 nm, and NIR–780 nm), and NDVI was calculated as (Gong et al., 2003):

Figure 1 Highlights of the design features of the push-wheel sensing cart.

(A) The rear wheels are installed with a steering mechanism to facilitate cart movement along a slightly curved row line as seen in a center-pivot field. (B) The sensor-mounting arm is fixed to the cart frame by two adjustable brackets, allowing flexible placement of the sensor-mounting arm along the width of the cart. (C) The smaller-sized tubings fitted into larger-size ones (D) to allow the width of the cart (i.e., the distance between the left and right side wheels) to be adjustable from 152 cm to 203 cm, a feature useful for measuring crops with different row spacings. Photo credit: Shane Sieckenius.

Figure 2 Configuration of sensors on the push-wheel sensing cart.

(A) Three Apogee radiometers to pointing at different angles to measure canopy temperature. (B) An ultra-sonic sensor to measure canopy height. (C) CR3000 data logger housed in an enclosure to record data from radiometers and the ultra-sonic height sensor. (D) ACS-430 Crop Circle sensor to measure canopy vegetation indices. (E) GeoSCOUT Datalogger to record Crop Circle sensor data. (F) Hemisphere 101 GPS receiver with the signal-splitting cable connecting to the CR3000 and GeoSCOUT dataloggers. (G) A detached ACS-Crop Circle sensor mounted atop a backpack sensing frame (along with the GeoSCOUT Datalogger) for measuring tall vegetation. Photo credit: Xuejun Dong.

(1) NDVI=ρ780−ρ670ρ780+ρ670,

where ρ780 and ρ670 represent reflectance at NIR and red band, respectively. To allow us standardized accurate customizable environmental data sampling, we utilized a Campbell Scientific CR3000 voltage data recorder including a CF memory card and with its CRBasic language data collection program execution functions. We subsequently acquired precise raw sensor potentials and their associated GPS coordinate strings, recorded to table structured digital records of 200 ms granularity (companies and products are not endorsed by the authors and are presented for educational purposes only). A full list of the CRBasic program driving the CR3000 datalogger is available as Computer Code S1.

Field data acquisition and collection

Field crops of wheat, corn, and cotton were grown under deficit- and full irrigation regimes in a center pivot field at the Texas A&M AgirLife Research and Extension Center at Uvalde, TX in the 2017–2018 growing season. The full irrigation was determined based on full-replenishment of the crop evapo-transpiration (ET) measured at the Research Center facility, while deficit irrigation was determined as 60% of the full irrigation. Diurnal changes in canopy NDVI and leaf water potentials for the three crops were measured on five clear days (see Table 1 for further information). Canopy NDVI for wheat and cotton was measured using the sensing cart, while that of corn was measured using the backpack sensing frame. Leaf water potential was measured using a PMS-615 Pressure Chamber (PMS Instrument Company, Albany, OR, USA), with the high pressure generated by pure nitrogen gas. For wheat, one transect encompassing a full and a deficit irrigation field was used, while for corn and cotton, a circular transect within a quarter of the center pivot field was used for the measurement. Within the circular transect, six equal divisions were delineated—each at least 30-m long, covering 15° angle in the circle and randomly assigned to receive either deficit or full irrigation treatment, with the six sections designated as six plots, i.e., deficit-1, full-1, full-2, deficit-2, deficit-3 and full-3, starting at the west side and ending at the east side of the circular path. On each of the five days selected for making field measurement, NDVI was scanned within the transect repeatedly once every hour from early morning to late afternoon. On each day, the datalogger was turned on before start of the first scan, and kept on until completing the day’s measurement in late afternoon. Each hour when NDVI was measured, 4–6 healthy and representative leaves of corn, cotton or wheat were chosen for measuring leaf water potentials (LWP) under field conditions (Fig. 3). For wheat, the flag leaves were selected to make the LWP measurement; for corn and cotton, the fourth leaf counting down from the fully expanded youngest leaf were used for the LWP measurement. To minimize water loss from the leaves enclosed in the chamber during pressurization, one moistened, 15-cm diameter filter paper was lined against the interior of the chamber wall and maintained moistened throughout the measurements.

Figure 3 Ground-based phenotyping tools as used in field conditions.

(A) Field map showing locations of field plots within a 50-acre center pivot field (Map data ©2018 Google). Corn and cotton were planted in the northeast quarter, which was subdivided into six pies at 15° intervals. Wheat was planted in the southwest quarter, with a segment delineated for NDVI scanning, covering a deficit- and full irrigated area. (B) A back-pack sensing frame equipped with an ACS-430 Crop Circle sensor and a GeoScout Datalogger was used to scan the corn plots; also shown is a pressure chamber being used to measure leaf water potential (Photo credit: Xuejun Dong). (C) A push-wheel sensing cart equipped with an ACS-430 Crop Circle sensor, three Apogee radiometers and one ultrasonic height sensor (Photo credit: Gongneng Feng).

Table 1 Summary of crop management of wheat, corn and cotton crops and the field physiological sampling activities in the 2017–2018 growing season.

Item	Wheat	Corn	Cotton	
Variety used	Gallagher	DKC64-69	DP1044	
Day of planting	11/27/2017	3/7/2018	4/9/2018	
Row spacing (cm)	19	102	102	
# Plants/m	150	5.7	11	
Precipitation received (mm)	126	89	121	
Irrigation applied (mm)	83/62†	330/229	362/273	
Day of harvesting	5/14/2018	8/1/2018	8/15/2018	
Yield (ka/ha)	4,712‡	7,912/5,746§	1,225/878¶	
NDVI/LWP measured	4/12/2018, 4/19/2018	6/7/2018, 6/11/2018	8/9/2018	
Min/max air temperature (°C)$	15.8/30.5, 17.4/25.1	24.8/35.0, 24.6/35.5	24.9/36.5	
Days after planting (DAP)	136, 143	92, 96	113	
Growth stage	Grain-filling	Grain-filling	70% Boll open	
Notes:

† Amount applied in the full and deficit irrigation treatment, respectively.

‡ Grain yield under full irrigation.

§ Grain yield under full and deficit irrigation treatment, respectively.

¶ Lint yield.

$ Hourly minimum/maximum temperature the ACS-430 Crop Circle sensor was exposed to during the measurement.

To document changes in green biomass during the growing season, leaf area index (LAI) for each of the crops was measured using a LI-3100 Area Meter (Li-Cor Inc., Lincoln, NE, USA) through destructive sampling. The measurement was made weekly or once every 2 weeks during the periods when there were significant changes in green biomass. For wheat, three random shoot samples were collected by harvesting a 20-cm long of a representative row from both deficit and full irrigation segments of the transect. For corn and cotton, one representative plant per plot was harvested to measure total leaf area on each measurement period. LAI was estimated based on total green leaf area per plant, row spacing and population density (shown in Table 1).

Data processing and statistical analysis

In this study, the data analysis is focused on the measured NDVI and its relationship with LWP. Over five days, about 1.3 million lines of data were recorded using the ground-based phenotyping tools on three crops. Some of the data lines were recorded while the sensor was in rest awaiting next round of scans. This was done in order to avoid the datalogger being turned off and back on repeatedly during the day, causing inadvertent errors. These unwanted readings during idle period, as well as other unwanted readings, were deleted during data processing according to recorded GPS coordinates, as well as the Coordinated Universal Time (UTC), in order to match the sensor data with field plots/treatments and time of measurement on different days. This initial data processing was done in Minitab (Version 17.3.1, 2016; Minitab Inc., State College, PA, USA). The datalogger recorded UTC time was translated into the Central Standard Time (CST) during the local day-light-savings time period by the relation CST = UTC − 5. The summarized data of NDVI and LWP were then further analyzed as described below.

Two methods were employed to conduct the least-squares regression analysis describing the relationship between measured values of NDVI (x) and leaf water potential (y) in the form

(2) Y=mX+c,

where Y and X represent adjusted/calculated values corresponding to measured y and x, m and c represent the slope and y-intercept of the model. 1. The first method is default to most statistical software packages where a common uncertainty in y is assumed, and the measurement errors in x are considered negligible as compared to those of y. Specifically, the regression analysis was conducted using GraphPad Prism 6 (Version 6.07 for Windows, 2015, GraphPad Software, San Diego, CA, USA). Statistical differences among and between the slopes and the y-intercepts of the linear regressions representing different treatments were compared using ANCOVA in Prism 6. The uncertainties in the coefficients m and c were calculated according to Taylor (1982) and implemented using a Minitab macro ‘lsq.mac’ (see Computer Code S2 and Dataset S1). The maximum relative errors in predicted LWP given a specific NDVI observation was estimated according to rules of error propagation (Taylor, 1982):

(3) δy|y¯|≤1|y¯|[|∂y∂m|δm+|∂y∂c|δc]=1|y¯|(|x¯|δm+δc),

where δy, δm, and δc are estimated uncertainties in y, m, and c, and x¯ and y¯ are the mean values of x and y, respectively. The first method is subsequently referred to as the method of OLS (Ordinary Least Squares).

2. The second method is based on the principle of weighted linear least-squares considering measurement uncertainties in both coordinates (York, 1966). In this method, the scheme of Deming (1943) was used to give different data points different weighting factors ω (xi) and ω (yi) that were defined as the inverse squares of the uncertainties δ(xi) and δ(yi) (i.e., ω (xi) = 1/δ2 (xi) and ω (yi) = 1/δ2 (yi)), and the best values of m and c of Eq. (2) were identified by minimizing the sum of the weighted squared residuals S as:

(4) S=∑i=1N[ω(xi)(xi−Xi)2+ω(yi)(yi−Yi)2],

where xi and yi are observed values and Xi and Yi and calculated ones with i running from 1 to N. York (1966), McIntyre et al. (1966) and Williamson (1968) developed the full solution of the dual-uncertainties least-squares problem, which has received wide recognition in geosciences and physical sciences (Cantrell, 2008). However, the same method appears to be less-utilized by biologists who may equally be faced with the same type of statistics problems as do geoscientists, physicists and chemists. For this reason, it is necessary to summarize the main steps for finding the model parameters and their uncertainties in this second method, based on equations of York (1969) and Reed (1992).

With the “overall weight” for the ith data point defined as

(5) Zi=ω(xi)ω(yi)m2ω(yi)+ω(xi),

Equation (4) can be written as

(6) S=∑i=1NZi(yi−mxi−c)2.

The pursuit of minimization of Eq. (6) leads to a “least-squares cubic” in the form of

(7) m3∑i=1NZi2Ui2ω(xi)−2m2∑i=1NZi2UiViω(xi)−m[∑i=1NZiUi2−∑i=1NZi2Vi2ω(xi)]+∑i=1NZiUiVi=0,

where Zi is from Eq. (5), Ui=xi−∑i=1NZixi(∑i=1NZi)−1 and Vi=yi−∑i=1NZiyi(∑i=1NZi)−1. Equation (7) can be solved for m to obtain the best value of the slope. However, since Zi, Ui and Vi are themselves all functions of m, the equation is actually a pseudo-least-squares cubic, and the best value of m is obtained through successive iterations starting from a crude estimation, such as from an unweighted least-squares fitting. A least-squares cubic similar to Eq. (7) with the associated iterative solution was developed independently by McIntyre et al. (1966) in analyzing the Rb-Sr isochrons from geological samples. Equation (7) may also be formulated in pseudo-quadratic- (York, 1969; Reed, 1992) or pseudo-linear form (Williamson, 1968; Reed, 2015), but they all need to be solved iteratively, similar to the cubic case. Using multiple data sets, Reed (1989) demonstrated that finding the correct solution to Eq. (7) can sometimes be tricky and confusing, and provided a direct searching method to quickly “pin down” the true value of m starting from a relatively good “seed” value. Once the best value of m is obtained, the like value of c can be found from

(8) c=∑i=1NZiyi(∑i=1NZi)−1−m∑i=1NZixi(∑i=1NZi)−1.

Built on the work of York (1966), Reed (1992) provided a corrected expression of the variance of m:

(9) δm2=SN−2∑i=1N[1ω(yi)(∂m∂yi)2+1ω(xi)(∂m∂xi)2],

where S is from Eq. (6), ∂m∂yi and ∂m∂xi can be computed from xi, yi, ω (xi), ω (yi) and the best value of m (see Appendix of Reed (1992) for the rather involved explicit expressions). The expression for δc2 (variance of c) is identical to that of δm2 but with derivative of c in place of m in Eq. (9).

Several researchers (McIntyre et al., 1966; York, 1969; Lybanon, 1984; Reed, 1989; Press et al., 1992), and a few others as discussed by Cantrell (2008), have developed computer programs (mainly in Fortran) to implement the algorithms of finding both the best parameter values (i.e., m and c) and their variances for the least-squares fitting problem with errors in both coordinates. Notably, Cantrell (2008) developed an Excel spreadsheet program to implement the Williamson-York method of bivariate linear fit. Reed (2010) translated his Fortran program into an easy-to-use Excel spreadsheet, and later updated it (Reed, 2015) to account for the situation with correlated x-y uncertainties, one that occurs widely in the study of isotope ratios in geosciences (York, 1969; York et al., 2004). However, in this paper, we only consider the situation with non-correlated x-y uncertainties, because the interested quantities (i.e., NDVI and LWP) were derived from measurements using different types of equipment independently, and thus their uncertainties are assumed to be uncorrelated.

Our second method relied on the use of the Excel spreadsheet ‘LLS(SIGMAS).xls’ of Reed (2010). Specifically, the Excel function ‘Goal seek’ was used to obtain the best estimation of coefficient m that ensured a function g(m) (such as Eq. (7)) taking the value of zero. Then the best value of m was used to determine the best value of c according to Eq. (8). To facilitate the determination of m, it was important to first use the unweighted least-squares routine builtin in ‘LLS(SIGMAS).xls’ to get a seed value of m which was then input to the ‘Goal seek’ function to find the best value of m (Reed, 2010). Since the values in both coordinates are adjusted in this method, the uncertainties δm, and δc can be calculated either at the observed points or at the calculated/adjusted points (Reed, 2010). In ‘LLS(SIGMAS).xls’, this is achieved using a switch: to evaluate the derivatives at the observed points, set PASS = 0, or to evaluate them at the calculated/adjusted points, set PASS = 1. We used the second option (i.e., at the adjusted points) but the differences in results using the two options were generally small for well-correlated data. Equipped with the best estimates of m and c and their variances (see Eq. (9)), which are part of the outputs of the above spreadsheet program, the maximum relative errors in predicted LWP using the second method were calculated according to Eq. (3). This method is subsequently referred to as the method of WLS (Weighted Least Squares).

We also used statistical resampling to investigate the effect of measurement errors in LWP on the regression model. In particular, we wanted to see how the reduced sample size of LWP would affect the estimated regression coefficients and their uncertainties. The resampling was done using a Minitab macro ‘sem_a.mac’ (see Computer Code S3 and Dataset S2), in which multiple sets of resampled 2 or 3 replicates of LWP values were drawn randomly (with replacement) from the original 4–6 measurements made during the field surveys. The resampled values of LWP were then used as inputs, along with the originally measured values of NDVI, to ‘lsq.mac’ and ‘LLS(SIGMAS).xls’, in order to obtain the best estimates of the regression coefficients and their uncertainties using the method of OLS and WLS, respectively. Differences in the regression coefficients (and associated uncertainties) between the situation with resampled LWP values and that with the originally measured LWP values were compared using the One-sample t-test available in Minitab.

Results

NDVI in relation to leaf area index and leaf water potential

Average values of LAI of wheat, corn and cotton were significantly lower under deficit irrigation than under full irrigation (p < 0.05, Fig. 4), especially in the mid- to late growth stages when canopy NDVI and leaf water potentials were measured. These trends in LAI paralleled those in measured NDVI, as seen in corn measured on June 7, 2018 (Fig. S1). In the case of the latter, it is evident that the differences between the deficit and full irrigation grew larger with the progression of time during a day. However, the NDVI values of wheat measured on April 12, 2018 exhibit a different trend, one in which the values measured from the deficit irrigation was high than that from the full irrigation, showing an opposite trend with the measured LWP (Fig. 5A). This was an unexpected result, but we did notice that during this day’s measurement there was significant lodging in the wheat field. This happened especially after the most recent irrigation event, possibly related to the Hessian fly infection. The measured NDVI values on April 19, however, show an expected trend, being higher in full than deficit irrigation (Fig. 5B). As a result, except for wheat measured on April 12, a higher LWP was associated with a higher NDVI in the three field crops.

Figure 4 Seasonal trends of leaf area index (LAI) for wheat (A), corn (B) and cotton (C) under full- and deficit irrigation regimes in 2018. LAI was estimated based on harvested sample plants.

For wheat, three representative row segments each 20-cm long were harvested in each irrigation treatment once every two weeks from February 1 (65 days after planting, DAP), and for corn and cotton, one plant was harvested in each of the three plots in each irrigation regime once every week starting from April 2 (27 DAP for corn) and May 22 (43 DAP for cotton). Error bars indicate standard errors of means.

Figure 5 Diurnal variations of leaf water potential (LWP) and canopy NDVI measured for wheat on April 12 and 19, 2018 (A, B), corn on June 7 and 11, 2018 (C, D), and cotton on August 9, 2018 (E) under full and deficit irrigation regimes.

Error bars indicate standard errors of means (no error bars shown for NDVI of wheat, since the plots were not replicated).

In drawing Fig. 5A, the NDVI values measured at 8 am (NDVI = 0.55 and NDVI = 0.63 for full and deficit irrigation, respectively; see raw data file NDVI_wheat_april_12.xls in the Supplemental Material) were not used, since these values were collected when the Crop Circle sensor was not fully warmed up in the relatively cold morning of April 12, 2018, with an air temperature as low as 15.8 °C at 8 am (see Table 1 and Fig. S2). The peculiar observation that the wheat NDVI under deficit irrigation was higher than that under full irrigation as measured on April 12 (Fig. 5) prompted us to inspect the raw data of canopy reflectance measured at the red and NIR bands. As seen in Fig. 6, the higher wheat NDVI on April 12 in the deficit irrigated plots was due both to a lower reflectance at the red band and the higher reflectance at the NIR band, as compared with the plots under full irrigation (see the definition of NDVI in Eq. (1)). The observation that the wheat NDVI in plots of deficit irrigation on April 19 was higher than that of the full irrigation plots was, however, due only to the higher red band reflectance associated with the deficit irrigation, because the NIR reflecance was similar between the two irrigation regimes (again see Fig. 6).

Figure 6 Box plots of mean reflectance values from 670 nm wavelength (A) and those from 780 nm wavelength (B) for measurements made in winter wheat plots at 10 times on April 12 (n = 10) and eight times on April 19 (n = 8).

Data are shown as the medians (central lines), interquartile range boxes, whiskers (representing the bottom 25% and top 25% of the data values), and an outlier (asterisk). Different lower-case letters “a–b” indicate significant difference (p < 0.0005) in mean reflectance values at a specific wavelength made on a particular day, while the same lower-case letters “a–a” indicate non-significant difference from the t-test (p > 0.05).

Another important point is if there was a significant difference in NDVI within a single day. For winter wheat, this was not tested statistically because the NDVI measurement was not replicated. By visual inspection (Figs. 5A, 5B), the diurnal trend of NDVI was unclear in both April 12 and 19, which was in contrast to the clear diurnal pattern of LWP in those two days. For corn and cotton, however, there were clear diurnal changes in NDVI (Figs. 5C, 5D, 5E). One-way ANOVA (n = 3, since each of the NDVI scans for corn or cotton was done with three replicates) indicated that, except for corn under full irrigation on June 11 that exhibited a marginally significant diurnal change (p = 0.056), five other diurnal courses of NDVI for corn/cotton all showed significant or highly significant diurnal trends (p < 0.05).

Since the relationship between LAI and NDVI was strong (Fig. S3), and there was significant differences in LAI values between deficit and full irrigation treatments, we analyzed the relationship between NDVI and LWP separately across the different crop and different irrigation treatments (Fig. 7). For wheat, there was an unclear or negative relationship between NDVI and LWP (Fig. 7A), while for corn and cotton, the relationship was positive. Though in the case of corn under full irrigation on June 11, 2018, the slope of the linear regression was not significantly different from zero (Figs. 7B, 7C). For those statistically significant regression lines shown in Fig. 7, further test indicated that, within corn or cotton, the slopes of the regressions lines are not significantly different, but the y-intercepts are significantly different (Table 2). Despite the low number of data points used in fitting the linear regression models, the relationships are generally strong as evidenced in the high R2 values in Table 2.

Figure 7 Relationship between leaf water potential and NDVI for wheat (A, B), corn (C, D) and cotton (E) based on measured values made under different irrigation regimes.

For wheat, the measurement was made on April 12 (135 DAP) and 19 (142 DAP), corn was on June 7 (93 DAP) and 11 (97 DAP), and cotton on August 9, 2019 (122 DAP). For a group of data in which the linear relationship was statistically significant, a best-fit line was drawn; otherwise, no line was displayed. Data values measured under full irrigation are shown as open circles, and those measured under deficit irrigation are shown as solid circles. Error bars indicate standard errors of means. Further information of the liner regressions is displayed in Table 2.

Table 2 Further information of statistically significant linear regressions describing leaf water potential as a function of NDVI as shown in Fig. 7.

Different uppercase letters indicate significant differences in regression slopes or y-intercepts for a particular crop type.

Crop	Date	Irrigation†	Slope	y-Intercept	P-value	R2	
Wheat	Apr. 12	F	−31.56	19.88	0.0001	0.93	
Corn	Jun. 7	D	29.55A	−21.71B	0.0001	0.89	
	Jun. 7	F	28.85A	−22.15C	0.0272	0.53	
	Jun. 11	D	23.18A	−16.40A	0.0145	0.60	
Cotton	Aug. 9	D	5.508A	−5.629A	0.0095	0.64	
	Aug. 9	F	5.051A	−5.708A	0.0117	0.62	
Note:

† “D” and “F” represent “deficit” and “full” irrigation.

Uncertainty analysis of NDVI and leaf water potential

Table 3 shows the uncertainties of the regression coefficients for the six significant linear regressions using the OLS method. In the last column of this table, the relative errors of the predicted LWP are shown. Since there was no statistical significance in regression coefficients between two cotton datasets (full vs. deficit irrigation), they were combined to form a larger dataset for further analysis. This is shown in the last row of Table 3. We can see that the prediction errors were large, with the best being 34% when the two cotton datasets were combined. However, when the uncertainties in both coordinates were considered in the least-squares regression analysis using the WLS method, the uncertainties in both coefficients were reduced, and also reduced were the prediction errors (Table 4). Further calculation details of the results in Tables 3 and 4 are shown in Dataset S3.

Table 3 Best estimates of the slopes (m) and y-intercepts (c), and associated uncertainties (δm and δc), for the significant linear regressions (assuming a common uncertainty in all y measurements).

The number of data points in each regression is indicated in column n, and the relative error for predicted leaf water potential based on uncertainties in m and c is shown in column δY/Y.

Crop	Date	Irrigation†	n	m (δm)	c (δc)	δY/Y	
Wheat	Apr. 12	F	8	−31.56 (3.63)	19.88 (2.52)	2.52	
Corn	Jun. 7	D	9	29.55 (3.90)	−21.71 (2.68)	4.53	
	Jun. 7	F	9	28.85 (10.37)	−22.15 (7.54)	12.82	
	Jun. 11	D	9	23.18 (7.18)	−16.40 (4.55)	5.19	
Cotton	Aug. 9	D	9	5.51 (1.56)	−5.63 (0.75)	0.50	
	Aug. 9	F	9	5.05 (1.49)	−5.71 (0.85)	0.60	
	Aug. 9	D/F	18	4.07 (0.93)	−5.04 (0.49)	0.34	
Note:

† “D” and “F” represent “deficit” and “full” irrigation.

Table 4 Best estimates of the slopes (m) and y-intercepts (c), and associated errors (δm and δc), for the significant linear regressions in Fig. 7 after taking into consideration the measurement errors of both leaf water potential and NDVI.

The number of data points in each regression is indicated in the column n, and the relative error for predicted leaf water potential based on uncertainties in m and c is shown in the column δY/Y.

Crop	Date	Irrigation†	n	m (δm)	c (δc)	δY/Y	
Wheat	Apr. 12	F	8	−28.95 (3.02)	18.03 (2.13)	2.10	
Corn	Jun. 7	D	9	31.91 (3.40)	−23.31 (2.32)	4.04	
	Jun. 7	F	9	42.17 (7.76)	−31.85 (5.65)	9.47	
	Jun. 11	D	9	67.45 (22.41)	−44.36 (14.35)	16.40	
Cotton	Aug. 9	D	9	4.75 (1.04)	−5.34 (0.51)	0.33	
	Aug. 9	F	9	4.88 (1.34)	−5.64 (0.73)	0.52	
	Aug. 9	D/F	18	4.24 (0.77)	−5.19 (0.39)	0.27	
Note:

† “D” and “F” represent “deficit” and “full” irrigation.

The difference between the results using two regression methods was further illustrated by the relative errors of the regression coefficients m and c. As seen in Table 5, the WLS method yielded lower relative sizes of the uncertainties for the coefficients, as compared with the OLS method, except for corn measured on June 11. The latter case was likely due to the very high relative errors in measured NDVI for corn on this day, as shown in Table 6.

Table 5 Percentage errors in m and c for the significant linear regressions using the ordinary least-squares (OLS) and weighted least-squares (WLS) methods.

Crop	Date	Irrigation†	n	δmm(%) OLS/WLS	δcc(%) OLS/WLS	
Wheat	Apr. 12	F	8	12/10	13/12	
Corn	Jun. 7	D	9	13/11	12/10	
	Jun. 7	F	9	36/18	34/18	
	Jun. 11	D	9	31/33	28/32	
Cotton	Aug. 9	D	9	28/22	13/10	
	Aug. 9	F	9	30/27	15/13	
	Aug. 9	D/F	18	23/18	10/8	
Note:

† “D” and “F” represent “deficit” and “full” irrigation.

Table 6 Measurement uncertainties (d) relative to measurement ranges (R) of NDVI and LWP (expressed as percentage) observed in different crops/days at Uvalde in 2018.

Crop	Date	Irrigation†	n	δNDVIRNDVI(%)	δLWPRLWP(%)	
Wheat	Apr. 12	F	8	–	6.0	
Corn	Jun. 7	D	9	10.5	3.8	
	Jun. 7	F	9	10.5	3.8	
	Jun. 11	D	9	19.8	3.7	
Cotton	Aug. 9	D	9	6.6	5.3	
	Aug. 9	F	9	6.4	5.9	
Notes:

† “D” and “F” represent “deficit” and “full” irrigation.

The 20 sets of resampled LWP values and the corresponding uncertainties for cases of 2 and 3 resampled values of LWP are shown in columns 1–40 of Datasets S4 and S5, respectively. The resultant model coefficients and their uncertainties after these resampled LWP values were used as inputs to the OLS procedure of ‘lsq.mac’ are shown in columns 41–44 of Datasets S4 and S5, respectively. The resultant model coefficients and their uncertainties using the WLS procedure of ‘LLS(SIGMAS).xls’ are shown in Datasets S6 and S7, respectively. The results of statistical resampling indicate that reducing the sample size of LWP to 2 or 3 significantly increased the uncertainties of the estimated coefficients m (0.92 vs. 0.77) and c (0.48 vs. 0.39) as compared with the respective uncertainties when the original 4 replicates of LWP were used as inputs to the WLS method (Fig. 8). Reducing the sample size of LWP also increased the prediction uncertainty of LWP to 0.32 (from 0.27 using the WLS method as seen in Table 4). However, there was no significant difference in the uncertainty of either coefficient when the OLS method was used, as seen by the dashed lines passing through or touching the error bars of the resampled values using the OLS method (Fig. 8). This result was anticipated since no errors from the individual observations of NDVI or LWP were considered in the OLS method.

Figure 8 Best estimates of regression coefficients m and c and their uncertainties using 20 random resamples of two or three replicates of cotton LWP drawn from the original four replicates.

Error bars indicate standard errors of means and dotted lines show the values of the regression coefficients, or their uncertainties, when errors of LWP were based on all four replicates of LWP using the method of ordinary least-squares (OLS) or weighted least-squares (WLS).

Discussion

Five out of the six significant linear regressions relating NDVI to LWP yielded positive slopes (Fig. 7 and Table 2), suggesting that an increased value of NDVI was associated with a less negative value of LWP. This trend was similar to that obtained in some other studies (Mastrorilli et al., 2010; Baluja et al., 2012; Maciel et al., 2020). In particular, the magnitudes of both the y-intercepts and slopes for cotton obtained in our study using the WLS method (see Table 4) were closely comparable with the respective values of an equation for coffee, LWP = −4.329 + 4.806 NDVI (R2 = 0.82), obtained by Maciel et al. (2020). The negative slope found in the well-watered wheat on April 12, 2018, however, was rather perplexing. We also had three datasets in which no significant linear relation between LWP and NDVI was established (see Figs. 7A, 7B). Furthermore, the NDVI values in plots of deficit irrigation being higher than those of full irrigation on this same date (April 12) was also unexpected. The higher reflectance at the red band for wheat at full irrigation (Fig. 6A) suggests that wheat leaves under full irrigation were stressed more than those under deficit irrigation, a trend opposite of that of the typical stress responses of leaf reflectance in the VIS band (Carter, 1993), in which an increased red band reflectance was observed due to reduced absorption from the stress-related reduction in photosynthetic pigments. The observed lodging related to the Hessian fly infection in our wheat plots may have hastened the senescence during this early grain-filling stage, especially for the full irrigation plots. This can be inferred from Fig. 4A, in which, toward end of the measurements, a faster reduction of LAI was observed in full irrigation plots than those in deficit irrigation. The above discussion casts doubt for the LWP-NDVI relation of wheat in Fig. 7 as being representative of the normal conditions.

Although all the six significant linear regressions in Table 2 showed high coefficients of determination, the uncertainties in both the estimated best model coefficients of m and c and in the predicted LWP were large (Tables 3 and 4), especially for corn and wheat. Importantly, taking into consideration the uncertainties of both LWP and NDVI led to reduced uncertainties for the model coefficients, as well as reduced upper bounds of the uncertainties in the predicted LWP using the average NDVI values (Tables 4 and 5, Dataset S3). One exception was found in the case of corn measured on June 11, which was likely due to the very high relative error in measured NDVI for corn (Table 6). A notable finding is that the prediction uncertainty for the combined cotton samples was reduced from 34% to 27% after the WLS method was used. To what extent this effect (of reduced prediction uncertainty associated with the use of WLS) also applies to a different context of biological data analysis is not intuitively evident just by inspecting the relevant equations for calculating the uncertainties of the model coefficients, such as Eq. (9). But our experience of using this method in our data analysis suggests a promising outcome, with implications to other studies involving linear regressions of experimentally measured variables. The WLS method we used in this study was due to Reed (2010), but the solution of the problem was the result of multiple efforts dating back to the work of York (1966) and others, and later of Reed (1989, 1992).

With the maximum prediction error being 27% for combined cotton samples, the equation developed in this study may only be able to separate the differences in LWP between the extreme crop varieties growing under field conditions as seen in an example shown in Fig. S4 (Dong & Mott, 2021) and also in Mart, Veneklaas & Bramley (2016). To increase the range of variation of NDVI when variables such as green biomass or leaf area index are controlled in order to detect the variability of LWP, it may be necessary to create a wide range of water availability, possibly including both well-watered and stressed treatments. In addition, stronger signals due to changes in LWP may be captured by sensors with wider NIR bands, preferably reaching 1,000 nm (Peñuelas & Filella, 1998; Elsayed, Mistele & Schmidhalter, 2011), yet a wider band may incur more noise outside the biological signal targeted. Further studies are needed to confirm if the above strategies may help to reduce the relative errors as depicted in Table 6. In particular, additional research is needed to understand why under some circumstances, such as in cotton in our study, the range of NDVI measurement was wider and the uncertainty relative to the measurement range was smaller than in other situations, such as in corn.

Using statistical resampling, we showed that the beneficial effect of reduced model uncertainties due to the use of the WLS method diminished if the sample size of LWP is reduced (Fig. 8). In the case of combined cottons samples, reducing the sample size of LWP from 4 to 2 or 3 increased the prediction uncertainty of the model from 27% to to 32%, a value closer to that obtained using the unweighted least-squares (34% as seen in Table 3). This result emphasized the importance of maintaining or not reducing the sample size of LWP while calibrating the LWP-NDVI equations, even though the field measurement of LWP is time-consuming using some of the traditional methods (Cheung, Tyree & Dainty, 1975; Bennett, Cortes & Lorens, 1986).

Since ACS-430 is an active sensor that has its own light source, the impact of changes in solar angle at different times of a day was assumed to be small, although Kim et al. (2012) observed some changes in canopy reflectance within a day. In addition, we assumed that the effects of changes in diurnal leaf movement (Wang et al., 2004; Greenham et al., 2015; Cal et al., 2018), leaf area shrinkage (Hilty, Pook & Leuzinger, 2009; Dong et al., 2011), as well as chlorophyll positioning (Maai et al., 2020), were minimum to our measured values of NDVI, and the key signals captured by our active spectral sensor, which was consistently positioned at a common height on the sensing frame looking down vertically at the plant canopy, originated mainly from the changes in leaf water status of the closed canopies of wheat, corn or cotton. We noted a moderate correlation between air temperature and NDVI for the combined corn/cotton samples (R2 = 0.21; see Fig. S5), which was in contrast to the strong correlation between LWP and NDVI in the combined data (R2 = 0.78). Our field experience also indicated that the Crop Circle sensor needs more time (perhaps at least 1 h) to warm up under a lower air temperature condition, such as that encountered in the early morning of April 12, 2018 when measuring wheat (Fig. S2). Apparently, more rigorous approaches involving controlled experiments are needed to fully resolve the impact of these potential confounding effects, and those of other environmental factors mentioned above, on accurate sensing of LWP remotely using spectral reflectance from plant canopies.

In this study, our analysis was focused on the use of NDVI to indicate LWP of selected crop plants, although a number of other vegetation indices can also be derived from the same sensor as used here. Using the same active sensor, Dong, Feng & Zemach (2021) showed that, similar to NDVI, the red band reflectance and re-normalized difference vegetation index (RDVI) explained >62% of the variation in shoot water content of sesame during drydown. Zarco-Tejada, González-Dugo & Berni (2012) found that RDVI was more strongly related to LWP than was NDVI in a citrus orchard (R2 = 0.44 vs. R2 = 0.24). Ramoelo et al. (2015) demonstrated that normalized water index (NDWI) explained 70% of LWP of selected plants in an indigenous vegetation in South Africa. These additional indices, as well as others, such as the normalized difference moisture index (NDMI) derived from Landsat (https://www.usgs.gov/core-science-systems/nli/landsat/normalized-difference-moisture-index), may be useful for further in-depth studies on the relationship between spectral indices and plant water status.

Another limitation of our study originates from the fact that NDVI was measured at the canopy level while LWP was from the leaf level. The extent to which this mis-match and the associated uncertainty may have affected a reasonable characterization of the canopy water status was not assessed in this study, since it was difficult to conduct a comprehensive survey of LWP distribution within the studied plant canopies using the pressure chamber method. Although the vertical gradient of LWP within canopies of most agricultural plants should be modest, considering the theoretical and measured gradient of about −0.01 MPa m−1 (Scholander et al., 1965; Bauerle et al., 1999), leaf water status can vary in relation to leaf age (Jordan, Brown & Thomas, 1975) and position (Bader et al., 2014) within plant canopies. One way of workaround is to measure both NDVI and LWP at the leaf level (Stimson et al., 2005). However, a lot of times, the work of remote sensing of plant water status is motivated by the needs to characterize crop genotypes under field conditions, and thus the data from the limited measurements using the pressure chamber method is often used as a canopy surrogate to correlate with remotely measured canopy features using ground based sensors (such as in this current study) and sensors on board an unmanned aerial vehicle (Baluja et al., 2012; Zarco-Tejada, González-Dugo & Berni, 2012) or satellite (Maciel et al., 2020). We are unaware of the availability of an experimental method that can readily be used to characterize the LWP profile of a full plant canopy. Yet there is hope to rely on continuously measured leaf turgor pressure from different positions of a plant canopy using a set of magnetic-based ZIM-probes (Bader et al., 2014; Bramley et al., 2015; Martínez-Gimeno et al., 2017). Notably, the fine resolution in the time scale of the ZIM-probes can complement the fine spatial resolution of the the aerial-based sensors, such as that used by Zarco-Tejada, González-Dugo & Berni (2012), to achieve an optimal calibration of leaf water status under field conditions.

Conclusions

Our study demonstrated that, NDVI, a well-known vegetation index frequently used to indicate vegetation biomass and LAI, can also be of value in estimating LWP of growing plants, if sufficient care is taken to minimize the confounding effects from other factors such as changes in LAI. In particular, we addressed a few important methodical issues regarding the uncertainties of the regression models relating NDVI to LWP, which have not been well-understood in current literature pertaining to both the ground-based, aerial and satellite remote sensing, to quantify LWP using NDVI. This was achieved through the use of a custom-built phenotyping cart, as well as the application of the OLS and WLS methods for statistical error analysis of the field data. Our data analyses helped provide a deeper understanding of the factors influencing the LWP-NDVI relationship using readily available, low-cost spectral sensors. Our results showed that obtaining a high value of coefficient of determination did not guarantee a high prediction precision in the regression model relating NDVI to LWP. To improve the precision of using NDVI to predict LWP, measurement uncertainties in both LWP and NDVI should be considered through the use of the WLS regression, and the sample size of LWP should not be reduced when using the traditional pressure chamber method. Our data also showed instances in which the LWP-NDVI relationship was not significant or even suspicious (such as in wheat); these need to be investigated further in future studies. Finally, the uncertainty relative to the measurement range of NDVI may need to be controlled or reduced. This may be achieved by using sensors with NIR bands that can capture stronger or more stable signals of leaf water variability, sampling uniform, closed canopies to prevent confounding effects of changes in LAI/green biomass and plant disease status from influencing the signals of NDVI, and targeting crop varieties growing under variable and stress conditions with their coincident LWP values. The methods used in this paper may be extended to future studies of remote sensing of LWP in which additional spectral indices will be evaluated and tested, and more high-throughput LWP measurement methods be developed to characterize the full canopies of field crops.

Supplemental Information

Supplemental Information 1 A list of computer codes.

Supplementary Code S1. A list of the CR3000 program to accommodate the current sensor installation on the phenotyping cart.

Supplementary Code S2. Minitab macro ‘lsq.mac’ for computing the uncertainties in y (dependent variable), c (y-intercept), and m (slope) of a linear regression in the form Y = mX + c using data pairs (x1, y1), …, (xN, yN) and assuming negligible errors in x.

Supplementary Code S3. Minitab macro ‘sem_a.mac’ for (a) generating 20 sets of leaf water potential (LWP) data and associated uncertainties (standard errors of the means, SEM) by randomly sampling 2 or 3 replicates from 18 groups of 4 values of the measured LWP, and (b) estimating the uncertainties of the coefficients of the linear regression Y = mX + c for the resampled LWP data, along with a fixed set of data of measured NDVI for cotton. Here Y represents adjusted values for the dependent variable y (LWP), X represents adjusted values for the independent variable x (NDVI; here X = x for the ordinary least-squares method), and m and c are coefficients to be determined from the experimental data.

Click here for additional data file.

Supplemental Information 2 A sample input dataset for Supplementary Code S2.

A sample input dataset for Supplementary Code S2 (‘lsq.mac’). Columns 1-4 must be NDVI (mean), LWP (mean), NDVI (SEM), and LWP (SEM), where ‘mean’ stands for mean values and SEM for standard errors of means, defined as standard deviation divided by square root of N, the sample size.

Click here for additional data file.

Supplemental Information 3 Input dataset for Supplementary Code S3.

Input dataset for Supplementary Code S3 (‘sem_a.mac’). Eighteen columns of LWP measurements (each from a particular hour of a day under different irrigation regimes) must be entered in c1-c18, and the average values of NDVI for the respective hours of the day must be entered in c19 (with 18 rows of data).

Click here for additional data file.

Supplemental Information 4 Best estimates of the regression coefficients and the associated uncertainties for the linear model.

Best estimates of the regression coefficients and the associated uncertainties for the linear model Y = mX + c, describing LWP as a function of NDVI for different crops measured on different days in 2018 using either the ordinary least-squares (OLS), or the weighted least-squares (WLS) method. In the OLS method, relevant equations of J. R. Taylor (1982, pages 153–159) were used, while in the WLS method, an Excel spreadsheet developed by B. C. Reed (2010) was used. Also shown are the relative prediction errors based on average values of LWP and NDVI for different regressions, as well as the relative errors of regression coefficients m and c (expressed as %). References: (1) J.R. Taylor. (1982). An Introduction to Error Analysis. University Science Books. Mill Valley, CA. (2) B.C. Reed. (2010). A spreadsheet for linear least-squares fitting with errors in both coordinates. Physics Education 45: 93–96.

Click here for additional data file.

Supplemental Information 5 Output results of the Minitab macro ‘sem_a.mac’.

Output results of the Minitab macro ‘sem_a.mac’ when 2 replicates of the values of LWP were drown randomly from the original 4 measurements. Columns 1 through 20 store the 20 sets of resampled LWP values, and columns 21-40 store the corresponding standard errors of the means of the resampled LWP values. Columns 41 through 44 store the best estimates of the regression coefficients (m and c) and their uncertainties using the resampled LWP values as inputs to the OLS method.

Click here for additional data file.

Supplemental Information 6 Additional output results of the Minitab macro ‘sem_a.mac’.

Output results of the Minitab macro ‘sem_a.mac’ when 3 replicates of the values of LWP were drown randomly from the original 4 measurements. Columns 1 through 20 store the 20 sets of resampled LWP values, and columns 21-40 store the corresponding standard errors of the means of the resampled LWP values. Columns 41 through 44 store the best estimates of the regression coefficients (m and c) and their uncertainties using the resampled LWP values as inputs to the OLS method.

Click here for additional data file.

Supplemental Information 7 Best estimates of the regression coefficients and the associated uncertainties for the linear model.

Best estimates of the regression coefficients (m and c) and the associated uncertainties for the linear model Y = mX + c, describing LWP as a function of NDVI for cotton, when 20 resamples of LWP with 2 replicates each were used as inputs to the weighted least-squares (WLS) method. The resampled LWP with 2 replicates were obtained from columns 1 through 40 from Supplementary Dataset S4. To run the WLS method, the average values and SEM’s of NDVI were also used as inputs but were not resampled.

Click here for additional data file.

Supplemental Information 8 More results on the best estimates of the regression coefficients and the associated uncertainties for the linear model.

Best estimates of the regression coefficients (m and c) and the associated uncertainties for the linear model Y = mX + c, describing LWP as a function of NDVI for cotton, when 20 resamples of LWP with 3 replicates each were used as inputs to the weighted least-squares (WLS) method. The resampled LWP with 3 replicates were obtained from columns 1 through 40 from Supplementary Dataset S5. To run the WLS method, the average values and SEM’s of NDVI were also used as inputs but were not resampled.

Click here for additional data file.

Supplemental Information 9 NDVI values of the corn canopies under full and deficit irrigation regimes at nine times on June 7, 2018.

Time series plots of scanned NDVI values of the corn canopies under full and deficit irrigation regimes (blue open and red solid circles, respectively) at nine times on June 7, 2018. The NDVI values presented in each of the sub-figures represent the complete scans done at a particular hour across the northeast quarter of the center pivot field (see Fig. 3A in the main text), covering three full and three deficit plots. Also in each sub-figure, the average NDVI values for the full and deficit plots are indicated along with dotted lines (values from full irrigation are always higher than those from deficit irrigation).

Click here for additional data file.

Supplemental Information 10 Justification for excluding the NDVI values measured from wheat plots at 8 am on April 12, 2018 from further analysis.

(A) NDVI scans obtained at 8 am were substantially lower than those obtained from 10 am, 12 pm, 2 pm, 4 pm and 6 pm when the cart was moved passing the deficit and full plots. (B) NDVI scans obtained at 9 am were similar in magnitude when compared with data values obtained from 11 am, 1 pm, 3 pm and 5 pm when the cart was moved passing the full and deficit plots after one hour of rest. (C) NDVI scans obtained between 8 am and 10 am from the wheat plots, including both the measurements made when the cart was moving across the wheat plots and those when it was at rest (i.e., stationary). Note that at 8 am and 10 am, the order of scans was from deficit to full plots, while at 9 am it was the opposite (i.e., from full to deficit plots). The dashed blue line indicates the “running average” values of NDVI, which were obtained as the 7th average signals (A7) from a discrete wavelet transform (DWT) to the first 65536 sensor records using the computer program of Dong et al. (2008). A major strength of DWT for signal analysis is that it preserves localized subtle and abrupt variations, while also capturing the overall signal trends over time. The horizontal cyan line indicates the mean value of the 65536 NDVI records. This shows that, following the 8 am measurement, the NDVI values kept increasing during the first stationary period until 9 am. After that, the values became stabilized, as seen during the stationary period from 9 am to 10 am. To further visualize the contrasts in NDVI values before and after the sensor stabilization was reached, the 7th average signals (A7 from DWT) were sampled at every 100th point for both the resting periods from 8-9 am and 9-10 am in the order the values were recorded, resulting in 219 and 293 sampled pairs of NDVI-time index values, respectively. As can be seen in panel (D), the NDVI values during the stationary period from 8 am to 9 am were more strongly positively correlated with the index number (blue circles) than they were for the period after stabilization (from 9 to 10 am; see the red squares). Reference: X. Dong, P. Nyren, B. Patton, A. Nyren, J. Richardson and T. Maresca, 2008. Wavelets for agriculture and biology: A tutorial with applications and outlook. BioScience 58: 445–453.

Click here for additional data file.

Supplemental Information 11 Relationship between NDVI and leaf area index (LAI) for wheat, corn and cotton on selected five days in 2018.

Relationship between NDVI and leaf area index (LAI) for wheat, corn and cotton based on field measured values made on five days in 2018. Values of NDVI are averaged from field scans done at 10 am local time, and values for LAI are averaged from measurements made within 1–2 days from NDVI scans, except for values of corn at June 7 (93 DAP), which were interpolated linearly based on measured LAI values made on May 29 (84 DAP) and June 12, 2018 (98 DAP). For corn and cotton values from the full and deficit irrigation are indicated separately, while for wheat, the values from full and deficit irrigation are averaged, since the treatment was not replicated. Error bars indicate ± standard errors of means.

Click here for additional data file.

Supplemental Information 12 Ranges of leaf osmotic potentials of ten cotton varieties measured in 2020 in Lytle and Taylor, TX.

Ranges of measured leaf osmotic potentials of ten cotton varieties planted in 2020 under irrigated and dryland conditions in Lytle (Irrigated) and Taylor (dryland), TX. Upper panel: dot plot for individual leaf samples. lower panel: dot plot for individual varieties. Reference of data source: X. Dong and D. A. Mott. (2021). Leaf osmotic potential and morphological traits of 43 cotton varieties growing in a rainfall gradient from southwest to central Texas. In: In Boyd, S., Huffman, M., Krogman, L., and Sarkissian, A., editors, Proceedings of the 2021 Beltwide Cotton Conferences. Pages 193-197, Virtual. National Cotton Council of America.

Click here for additional data file.

Supplemental Information 13 NDVI in relation to leaf water potential (LWP) and air temperature (°C) for corn and cotton samples measured in 2018.

A comparison of the relationship between NDVI and leaf water potential (LWP) and that of the NDVI and air temperature (°C) for combined data of corn and cotton measured in 2018.

Click here for additional data file.

We thank Sixto Silva at the Texas A&M AgriLife Research and Extension Center at Uvalde for fabricating the frame structure of the push-wheel phenotyping cart and the back-pack sensing frame. We also appreciate Ray King for managing the field crops. We are grateful to the editor and three peer reviewers for insightful suggestions that helped to improve this paper. We thank Jennifer Dong for editing the main text. X.D. would like to dedicate this work to the memory of his M.S. and Ph.D. advisor, Prof. Xinshi Zhang (1934-2020).

Additional Information and Declarations

Competing Interests

Author Contributions

Data Availability

The authors declare that they have no competing interests.

Xuejun Dong conceived and designed the experiments, performed the experiments, analyzed the data, prepared figures and/or tables, authored or reviewed drafts of the paper, secured funding support, and approved the final draft.

Bin Peng performed the experiments, authored or reviewed drafts of the paper, secured funding support, and approved the final draft.

Shane Sieckenius performed the experiments, prepared figures and/or tables, authored or reviewed drafts of the paper, and approved the final draft.

Rahul Raman performed the experiments, prepared figures and/or tables, authored or reviewed drafts of the paper, and approved the final draft.

Matthew M. Conley conceived and designed the experiments, performed the experiments, analyzed the data, authored or reviewed drafts of the paper, and approved the final draft.

Daniel I. Leskovar conceived and designed the experiments, authored or reviewed drafts of the paper, secured funding support, and approved the final draft.

The following information was supplied regarding data availability:

Data is available at Zenodo:

Dong, Xuejun, Peng, Bin, Sieckenius, Shane, Raman, Rahul, Conley, Matthew M., & Leskovar, Daniel I. (2021). Data of leaf water potential, vegetation indices and leaf area indices of corn, cotton and wheat measured under field conditions [Data set]. Zenodo. DOI 10.5281/zenodo.4895650.

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
