# Peer review of "Leaf water potential of field crops estimated using NDVI in ground-based remote sensing—opportunities to increase prediction precision"

_PeerJ, doi:10.7717/peerj.12005_

## Round 0.1 · original submission · Major Revisions

It has been reviewed by experts in the field and we request that you make major revisions before it is processed further.

Reviewer 1 ·

Basic reporting

The study focused on the relationship between leaf water potential and normalized deficit vegetation index. The experiment is overall reasonable. However, there are still issues that need to be improved. The English writing must be improved before publishing since the current version is not professional. For the research, it is not clear how the WLS method could help with the analysis of the uncertainty. Please introduce more about this method because this part is important in the manuscript.

Detailed comments and suggestions:
L44 Misunderstanding statement
L51 In this paragraph, you mentioned NDVI could not represent LWP directly. So explain more how Elsayed did to exclude the effect.
L56 Following the previous comment, given the non-direct relationship with LWP and NDVI, what is the theoretic basis of the research in L57 to 58? Please read deeper in these studies, and give a more accurate review of LWP-NDVI relationship. Try to review these works considering their time scale.
L61-62. Unclear statement.
L83. Use “apply” or “improve” instead of “design”
L98. Delete the NDRE, since this part didn’t appear in the main result.
L106. Check the whole manuscript and make sure the phrase “Deficit- and Full irrigation” are the same (see L115 for the difference). And lowercase the initials of Deficit and Full.
L107. What is “A&M”
L123. Is there any difference between the top leaves of the canopy and lower leaves? If the answer is yes, how did you consider this uncertainty during analysis?
L157. As the major comments, please introduce the second method more to clarify how this method could describe the uncertainties.
L185. I strongly suggest dividing the result part into 2 part, to introduce the relationship and uncertainty analysis separately.
Table 1. Why there is no difference between full and deficit irrigated wheat?
L193. Check the manuscript, and avoid using the words like “surprising”.
L197. Analysis more why deficit irrigated wheat has a higher NDVI than full irrigated.
Figure 5. This figure is quite confusing. 1. Why there is a significant curve pattern within the first blue stage? 2. I think you just want to show the difference between full and deficit-irrigated NDVI, consider using a boxplot. 3. I am quite interested if there is a significant difference between NDVI within a single day? If the answer is yes, introduce is you are using a daily average.
Figure 7. Why use a linear fitting rather than using Beer-Lambert Law
Figure 8. Describe in chapter 2 more about the three dates. Describe the phenology of the crop at three times and the nearest irrigation event, to see if the unsatisfied result in panel a comes from irrigation.

Experimental design

No comment.

Validity of the findings

No comment.

Additional comments

No comment

·

Basic reporting

The manuscript is well drafted and structured.
All figures, tables and graphs are relevant and of good quality.
Authors have submitted the associated data, codes and supplemental materials.
The language and overall content is well written for international readers.

Experimental design

Overall methodology and design of research is well laid and conducted.
Authors have explained all steps in detail along with any notes that were considered while capturing the data.
All steps and methods are well supported by graphs and figures.

Validity of the findings

Its a good research that will definitely add to the existing knowledge in this field.
This is a ground research based on actual survey that means a lot more than simply gathering data and analyzing it.
Authors have well justified their works with proper discussion and results.

Reviewer 3 ·

Basic reporting

In this paper, a push-wheel cart was designed to collect field data and then the authors evaluated the potential of using NDVI as a proxy to calculate leaf water potential (LWP). Generally, the paper is well-constructed and I have some concerns about data collection and analysis.

Experimental design

The experiments are well designed as this may be the start point of a series of works. I suggest testing more variables as proxies and extend the work to the canopy level when calculating LWP using remote sensing data in future work.

Validity of the findings

It's fine.

Additional comments

1. The canopy reflectance was collected to calculate NDVI, so that the NDVI used in this study is the canopy NDVI, while, the LWP used in the study is the leaf LWP, are these two variables match? As we know, the canopy is affected by the Leaf area index (LAI), leaf angle distribution (LAD), surface soil reflectance, and the fraction of vegetation cover (FVC), etc. So, could you comment on the effect of canopy characters on your LWP-NDVI correlations? And have you considered using leaf NDVI and LWP to analyze the correlation? Or is possible to collect canopy LWP and analyze the correlation of NDVI---LWP at the canopy level? This may be interesting as we may using remote sensing data to analyze canopy level LWP in this way.
2. I noticed that both NDVI and NDRE were calculated for the evaluation, and how about the performance of NDRE? Furthermore, there are some other indices that could be used to describe crop water stress such as the NDWI (https://www.usgs.gov/core-science-systems/nli/landsat/normalized-difference-moisture-index), I suggest considering more indices in your further work.

---

## Round 0.2 · Minor Revisions

Please revise your manuscript accordingly.

Reviewer 1 ·

Basic reporting

no comment'

Experimental design

no comment'

Validity of the findings

no comment'

Additional comments

1. The date information should also be added in the description of Figure 5.
2. Authors should have another round of grammar check to make sure the English language satisfies the requirement of the journal.

Reviewer 3 ·

Basic reporting

I am satisfied with the revision.

Experimental design

The experiments are well designed.

Validity of the findings

It's fine.

Additional comments

I am satisfied with the revision.

---

## Round 0.3 · accepted · Accept

Thanks for the efforts to improve the manuscript. I consider this manuscript is ready for publication.

Reviewer 1 ·

Basic reporting

Satisfied

Experimental design

Satisfied

Validity of the findings

Satisfied

Additional comments

No more comments